# Elementary Symmetric Polynomials for Optimal Experimental Design

**Zelda Mariet**
Massachusetts Institute of Technology
Cambridge, MA 02139
zelda@csail.mit.edu

**Suvrit Sra**
Massachusetts Institute of Technology
Cambridge, MA 02139
suvrit@mit.edu

## Abstract

We revisit the classical problem of optimal experimental design (OED) under a new mathematical model grounded in a geometric motivation. Specifically, we introduce models based on elementary symmetric polynomials; these polynomials capture "partial volumes" and offer a graded interpolation between the widely used A-optimal design and D-optimal design models, obtaining each of them as special cases. We analyze properties of our models, and derive both greedy and convex-relaxation algorithms for computing the associated designs. Our analysis establishes approximation guarantees on these algorithms, while our empirical results substantiate our claims and demonstrate a curious phenomenon concerning our greedy method. Finally, as a byproduct, we obtain new results on the theory of elementary symmetric polynomials that may be of independent interest.

## 1 Introduction

Optimal Experimental Design (OED) develops the theory of selecting experiments to perform in order to estimate a hidden parameter as well as possible. It operates under the assumption that experiments are costly and cannot be run as many times as necessary or run even once without tremendous difficulty [33]. OED has been applied in a large number of experimental settings [35, 9, 28, 46, 36], and has close ties to related machine-learning problems such as outlier detection [15, 22], active learning [19, 18], Gaussian process driven sensor placement [27], among others.

We revisit the classical setting where each experiment depends linearly on a hidden parameter $\theta \in \mathbb{R}^m$. We assume there are $n$ possible experiments whose outcomes $y_i \in \mathbb{R}$ can be written as

$$y_i = x_i^\top \theta + \epsilon_i \quad 1 \le i \le n,$$

where the $x_i \in \mathbb{R}^m$ and $\epsilon_i$ are independent, zero mean, and homoscedastic noises. OED seeks to answer the question: how to choose a set $S$ of $k$ experiments that allow us to estimate $\theta$ without bias and with minimal variance?

Given a feasible set $S$ of experiments (i.e., $\sum_{i \in S} x_i x_i^\top$ is invertible), the Gauss-Markov theorem shows that the lowest variance for an unbiased estimate $\hat{\theta}$ satisfies $\mathrm{Var}[\hat{\theta}] = (\sum_{i \in S} x_i x_i^\top)^{-1}$. However, $\mathrm{Var}[\hat{\theta}]$ is a matrix, and matrices do not admit a total order, making it difficult to compare different designs. Hence, OED is cast as an optimization problem that seeks *an optimal design $S^*$*

$$S^* \in \underset{S \in [n], |S| \le k}{\operatorname{argmin}} \Phi\Big(\big(\sum_{i \in S} x_i x_i^\top\big)^{-1}\Big), \tag{1.1}$$

where $\Phi$ maps positive definite matrices to $\mathbb{R}$ to compare the variances for each design, and may help elicit different properties that a solution should satisfy, either statistical or structural.

Elfving [16] derived some of the earliest theoretical results for the linear dependency setting, focusing on the case where one is interested in reconstructing a predefined linear combination of the

underlying parameters $c^\top \theta$ (C-optimal design). Kiefer [26] introduced a more general approach to OED, by considering matrix means on positive definite matrices as a general way of evaluating optimality [33, Ch. 6], and Yu [48] derived general conditions for a map $\Phi$ under which a class of multiplicative algorithms for optimal design has guaranteed monotonic convergence.

Nonetheless, the theory of OED branches into multiple variants of (1.1) depending on the choice of $\Phi$, among which A-optimal design ($\Phi = $ trace) and D-optimal design ($\Phi = $ determinant) are probably the two most popular choices. Each of these choices has a wide range of applications as well as statistical, algorithmic, and other theoretical results. We refer the reader to the classic book [33], which provides an excellent overview and introduction to the topic; see also the summaries in [1, 35].

For A-optimal design, recently Wang et al. [44] derived greedy and convex-relaxation approaches; [11] considers the problem of constrained adaptive sensing, where $\theta$ is supposed sparse. D-optimal design has historically been more popular, with several approaches to solving the related optimization problem [17, 38, 31, 20]. The dual problem of D-optimality, Minimum Volume Covering Ellipsoid (MVCE) is also a well-known and deeply studied optimization problem [3, 34, 43, 41, 14, 42]. Experimental design has also been studied in more complex settings: [8] considers Bayesian optimal design; under certain conditions, non-linear settings can be approached with linear OED [13, 25].

Due to the popularity of A- and D-optimal design, the theory surrounding these two sub-problems has diverged significantly. However, both the trace and the determinant are special cases of fundamental spectral polynomials of matrices: *elementary symmetric polynomials (ESP)*, which have been extensively studied in matrix theory, combinatorics, information theory, and other areas due to their importance in the theory of polynomials [24, 30, 21, 6, 23, 4].

These considerations motivate us to derive a broader view of optimal design which we call *ESP-Design*, where $\Phi$ is obtained from an elementary symmetric polynomial. This allows us to consider A-optimal design and D-optimal design as special cases of ESP-design, and thus treat the entire ESP-class in a unified manner. Let us state the key contributions of this paper more precisely below.

**Contributions**

- We introduce ESP-design, a new, general framework for OED that leverages geometric properties of positive definite matrices to interpolate between A- and D-optimality. ESP-design offers an intuitive setting in which to gradually scale between A-optimal and D-optimal design.
- We develop a convex relaxation as well as greedy algorithms to compute the associated designs. As a byproduct of our convex relaxation, we prove that ESPs are geodesically log-convex on the Riemannian manifold of positive definite matrices; this result may be of independent interest.
- We extend a result of Avron and Boutsidis [2] on determinantal column-subset selection to ESPs; as a consequence we obtain a greedy algorithm with provable optimality bounds for ESP-design.

Experiments on synthetic and real data illustrate the performance of our algorithms and confirm that ESP-design can be used to obtain designs with properties that scale between those of both A- and D-optimal designs, allowing users to tune trade-offs between their different benefits (e.g. predictive error, sparsity, etc.). We show that our greedy algorithm generates designs of equal quality to the famous Fedorov exchange algorithm [17], while running in a fraction of the time.

## 2 Preliminaries

We begin with some background material that also serves to set our notation. We omit proofs for brevity, as they can be found in standard sources such as [6].

We define $[n] \triangleq \{1, 2, \ldots, n\}$. For $S \subseteq [n]$ and $M \in \mathbb{R}^{n \times m}$, we write $M_S$ the $|S| \times m$ matrix created by keeping only the rows of $M$ indexed by $S$, and $M[S|S']$ the submatrix with rows indexed by $S$ and columns indexed by $S'$; by $x_{(i)}$ we denote the vector $x$ with its $i$-th component removed. For a vector $v \in \mathbb{R}^m$, the *elementary symmetric polynomial (ESP)* of order $\ell \in \mathbb{N}$ is defined by

$$e_\ell(v) \triangleq \sum\nolimits_{1 \leq i_1 < \ldots < i_\ell \leq m} \prod\nolimits_{j=1}^{\ell} v_{i_j} = \sum\nolimits_{I \subseteq [m], |I| = \ell} \prod\nolimits_{j \in I} v_j, \qquad (2.1)$$

where $e_\ell \equiv 0$ for $\ell = 0$ and $\ell > m$. Let $\mathbb{S}_m^+$ ($\mathbb{S}_m^{++}$) be the cone of positive semidefinite (positive definite) matrices of order $m$. We denote by $\lambda(M)$ the eigenvalues (in decreasing order) of a symmetric matrix $M$. Def. (2.1) extends to matrices naturally; ESPs are *spectral functions*, as we set

$E_\ell(M) \triangleq e_\ell \circ \lambda(M)$; additionally, they enjoy another representation that allows us to interpret them as "partial volumes", namely,

$$E_\ell(M) = \sum\nolimits_{S \subseteq [n], |S| = \ell} \det(M[S|S]). \tag{2.2}$$

The following proposition captures basic properties of ESPs that we will require in our analysis.

**Proposition 2.1.** *Let $M \in \mathbb{R}^{m \times m}$ be symmetric and $1 \le \ell \le m$; also let $A, B \in \mathbb{S}_m^+$. We have the following properties: (i) If $A \succeq B$ in Löwner order, then $E_\ell(A) \ge E_\ell(B)$; (ii) If $M$ is invertible, then $E_\ell(M^{-1}) = \det(M^{-1})E_{m-\ell}(M)$; (iii) $\nabla e_\ell(\lambda) = [e_{\ell-1}(\lambda_{(i)})]_{1 \le i \le m}$.*

# 3 ESP-design

A-optimal design uses $\Phi \equiv \mathrm{tr}$ in (1.1), and thus selects designs with low average variance. Geometrically, this translates into selecting confidence ellipsoids whose bounding boxes have a small diameter. Conversely, D-optimal design uses $\Phi \equiv \det$ in (1.1), and selects vectors that correspond to the ellipsoid with the smallest volume; as a result it is more sensitive to outliers in the data[1]. We introduce a natural model that scales between A- and D-optimal design. Indeed, by recalling that both the trace and the determinant are special cases of ESPs, we obtain a new model as fundamental as A- and D-optimal design, while being able to interpolate between the two in a graded manner.

Unless otherwise indicated, we consider that we are selecting experiments *without* repetition.

## 3.1 Problem formulation

Let $X \in \mathbb{R}^{n \times m}$ ($m \ll n$) be a design matrix with full column rank, and $k \in \mathbb{N}$ be the budget ($m \le k \le n$). Define $\Gamma_k = \{S \subseteq [n] \text{ s.t. } |S| \le k, X_S^\top X_S \succ 0\}$ to be the set of *feasible* designs that allow unbiased $\theta$ estimates. For $\ell \in \{1, \ldots, m\}$, we introduce the *ESP-design* model:

$$\min_{S \in \Gamma_k} \quad f_\ell(S) \triangleq \tfrac{1}{\ell} \log E_\ell\big((X_S^\top X_S)^{-1}\big). \tag{3.1}$$

We keep the $1/\ell$-factor in (3.1) to highlight the homogeneity ($E_\ell$ is a polynomial of degree $\ell$) of our design criterion, as is advocated in [33, Ch. 6].

For $\ell = 1$, (3.1) yields A-optimal design, while for $\ell = m$, it yields D-optimal design. For $1 < \ell < m$, ESP-design interpolates between these two extremes. Geometrically, we may view it as seeking an ellipsoid with the smallest average volume for $\ell$-dimensional slices (taken across sets of size $\ell$). Alternatively, ESP-design can be also be interpreted as a regularized version of D-optimal design via Prop. 2.1-*(ii)*. In particular, for $\ell = m - 1$, we recover a form of regularized D-optimal design:

$$f_{m-1}(S) = \tfrac{1}{m-1}\big[\log \det\big((X_S^\top X_S)^{-1}\big) + \log \|X_S\|_2^2\big].$$

(3.1) is a known hard combinatorial optimization problem (in particular for $\ell = m$), which precludes an exact optimal solution. However, its objective enjoys remarkable properties that help us derive efficient algorithms for its approximate solution. The first one of these is based on a natural convex relaxation obtained below.

## 3.2 Continuous relaxation

We describe below a traditional approach of relaxing (3.1) by relaxing the constraint on $S$, allowing elements in the set to have fractional multiplicities. The new optimization problem takes the form

$$\min_{z \in \Gamma_k^c} \tfrac{1}{\ell} \log E_\ell\Big((X^\top \mathrm{Diag}(z)X)^{-1}\Big), \tag{3.2}$$

where we $\Gamma_k^c$ denotes the set of vectors $\{z \in \mathbb{R}^n \mid 0 \le z_i \le 1\}$ such that $X^\top \mathrm{Diag}(z)X$ remains invertible and $\mathbf{1}^\top z \le k$. The following is a direct consequence of Prop 2.1-*(i)*:

**Proposition 3.1.** *Let $z^*$ be the optimal solution to (3.2). Then $\|z^*\|_1 = k$.*

Convexity of $f_\ell$ on $\Gamma_k^c$ (where by abuse of notation, $f_\ell$ also denotes the continuous relaxation in (3.2)) can be obtained as a consequence of [32]; however, we obtain it as a corollary Lemma 3.3, which shows that $\log E_\ell$ is geodesically convex; this result seems to be new, and is *stronger* than convexity of $f_\ell$; hence it may be of independent interest.

**Definition 3.2** (geodesic-convexity). A function $f : \mathbb{S}_m^{++} \to \mathbb{R}$ defined on the Riemannian manifold $\mathbb{S}_m^{++}$ is called *geodesically convex* if it satisfies

$$f(P\#_t Q) \leq (1-t)f(P) + tf(Q), \qquad t \in [0,1], \text{ and } P, Q \succ 0.$$

where we use the traditional notation $P\#_t Q := P^{1/2}(P^{-1/2}QP^{-1/2})^t P^{1/2}$ to denote the *geodesic* between $P$ and $Q \in \mathbb{S}_m^{++}$ under the Riemannian metric $g_P(X, Y) = \text{tr}(P^{-1}XP^{-1}Y)$.

**Lemma 3.3.** *The function $E_\ell$ is geodesically log-convex on the set of positive definite matrices.*

**Corollary 3.4.** *The map $M \mapsto E_\ell^{1/\ell}((X^\top M X)^{-1})$ is log-convex on the set of PD matrices.*

For further details on the theory of geodesically convex functions on $\mathbb{S}_m^+$ and their optimization, we refer the reader to [40]. We prove Lemma 3.3 and Corollary 3.4 in Appendix A.

From Corollary 3.4, we immediately obtain that (3.2) is a convex optimization problem, and can therefore be solved using a variety of efficient algorithms. Projected gradient descent turns out to be particularly easy to apply because we only require projection onto the intersection of the cube $0 \leq z \leq 1$ and the plane $\{z \mid z^\top \mathbf{1} = k\}$ (as a consequence of Prop 3.1). Projection onto this intersection is a special case of the so-called continuous quadratic knapsack problem, which is a very well-studied problem and can be solved essentially in linear time [10, 12].

**Remark 3.5.** The convex relaxation remains log-convex when points can be chosen with multiplicity, in which case the projection step is also significantly simpler, requiring only $z \geq 0$.

We conclude the analysis of the continuous relaxation by showing a bound on the support of its solution under some mild assumptions:

**Theorem 3.6.** *Let $\phi$ be the mapping from $\mathbb{R}^m$ to $\mathbb{R}^{m(m+1)/2}$ such that $\phi(x) = (\xi_{ij}x_i x_j))_{1 \leq i,j \leq m}$ with $\xi_{ij} = 1$ if $i = j$ and 2 otherwise. Let $\tilde{\phi}(x) = (\phi(x), 1)$ be the affine version of $\phi$. If for any set of $m(m+1)/2$ distinct rows of $X$, the mapping under $\tilde{\phi}$ is independent, then the support of the optimum $z^*$of (3.2) satisfies $\|z^*\|_0 \leq k + \frac{m(m+1)}{2}$.*

The proof is identical to that of [44, Lemma 3.5], which shows such a result for A-optimal design; we relegate it to Appendix B.

# 4   Algorithms and analysis

Solving the convex relaxation (3.2) does not directly provide a solution to (3.1); first, we must round the relaxed solution $z^* \in \Gamma_k^c$ to a discrete solution $S \in \Gamma_k$. We present two possibilities: (i) rounding the solution of the continuous relaxation (§4.1); and (ii) a greedy approach (§4.2).

## 4.1   Sampling from the continuous relaxation

For conciseness, we concentrate on sampling without replacement, but note that these results extend with minor changes to with replacement sampling (see [44]). Wang et al. [44] discuss the sampling scheme described in Alg. 1) for A-optimal design; the same idea easily extends to ESP-design. In particular, Alg. 1, applied to a solution of (3.2), provides the same asymptotic guarantees as those proven in [44, Lemma 3.2] for A-optimal design.

---
**Algorithm 1:** Sample from $z^*$

---
**Data**: budget $k$, $z^* \in \mathbb{R}^n$
**Result**: $S$ of size $k$
$S \leftarrow \emptyset$
**while** $|S| < k$ **do**
    Sample $i \in [n] \setminus S$ uniformly at random
    Sample $x \sim \text{Bernoulli}(z_i^*)$
    **if** $x = 1$ **then** $S \leftarrow S \cup \{i\}$
**return** $S$

---

**Theorem 4.1.** *Let $\Sigma_* = X^\top \text{Diag}(z^*)X$. Suppose $\|\Sigma_*^{-1}\|_2 \kappa(\Sigma_*)\|X\|_\infty^2 \log m = \mathcal{O}(1)$. The subset $S$ constructed by sampling as above verifies with probability $p = 0.8$*

$$E_\ell\left(\left(X_S^\top X_S\right)^{-1}\right)^{1/\ell} \leq \mathcal{O}(1) \cdot E_\ell\left(\left(X_{S^*}^\top X_{S^*}\right)^{-1}\right)^{1/\ell}.$$

Theorem 4.1 shows that under reasonable conditions, we can probabilistically construct a good approximation to the optimal solution in linear time, given the solution $z^*$ to the convex relaxation.

## 4.2 Greedy approach

In addition to the solution based on convex relaxation, ESP-design admits an intuitive greedy approach, despite not being a submodular optimization problem in general. Here, elements are removed one-by-one from a base set of experiments; this greedy removal, as opposed to greedy addition, turns out to be much more practical. Indeed, since $f_\ell$ is not defined for sets of size smaller than $k$, it is hard to greedily add experiments to the empty set and then bound the objective function after $k$ items have been added. This difficulty precludes analyses such as [45, 39] for optimizing non-submodular set functions by bounding their "curvature".

---

**Algorithm 2:** Greedy algorithm

---

**Data**: matrix $X$, budget $k$, initial set $S_0$
**Result**: $S$ of size $k$
$S \leftarrow S_0$
**while** $|S| > k$ **do**
    Find $i \in S$ such that $S \setminus \{i\}$ is feasible and $i$ minimizes $f_\ell(S \setminus \{i\})$
    $S \leftarrow S \setminus \{i\}$
**return** $S$

---

Bounding the performance of Algorithm 2 relies on the following lemma.

**Lemma 4.2.** *Let $X \in \mathbb{R}^{n \times m} (n \geq m)$ be a matrix with full column rank, and let $k$ be a budget $m \leq k \leq n$. Let $S$ of size $k$ be subset of $[n]$ drawn with probability $\mathcal{P} \propto \det(X_S^\top X_S)$. Then*

$$\mathbb{E}_{S \sim \mathcal{P}}\left[ E_\ell\left( \left(X_S^\top X_S\right)^{-1}\right)\right] \leq \prod_{i=1}^{\ell} \frac{n-m+i}{k-m+i} \cdot E_\ell\left( \left(X^\top X\right)^{-1}\right), \tag{4.1}$$

*with equality if $X_S^\top X_S \succ 0$ for all subsets $S$ of size $k$.*

Lemma 4.2 extends a result from [2, Lemma 3.9] on column-subset selection via volume sampling to all ESPs. In particular, it follows that removing one element (by volume sampling a set of size $n-1$) will in expectation decrease $f$ by a multiplicative factor which is clearly also attained by a greedy minimization. This argument then entails the following bound on Algorithm 2's performance. Proofs of both results are in Appendix C.

**Theorem 4.3.** *Algorithm 2 initialized with a set $S_0$ of size $n_0$ produces a set $S^+$ of size $k$ such that*

$$E_\ell\left( \left(X_{S^+}^\top X_{S^+}\right)^{-1}\right) \leq \prod_{j=1}^{\ell} \frac{n_0-m+j}{k-m+j} \cdot E_\ell\left( \left(X_{S_0}^\top X_{S_0}\right)^{-1}\right) \tag{4.2}$$

As Wang et al. [44] note regarding A-optimal design, (4.2) provides a trivial optimality bound on the greedy algorithm when initialized with $S_0 = \{1, \ldots, n\}$ ($S^*$ denotes the optimal set):

$$E_\ell\left( \left(X_{S^+}^\top X_{S^+}\right)^{-1}\right)^{1/\ell} \leq \frac{n-m+\ell}{k-m+1} f(\{1,\ldots,n\}) \leq \frac{n-m+\ell}{k-m+1} E_\ell\left( \left(X_{S^*}^\top X_{S^*}\right)^{-1}\right)^{1/\ell} \tag{4.3}$$

However, this naive initialization can be replaced by the support $\|z^*\|_0$ of the convex relaxation solution; in the common scenario described by Theorem 3.6, we then obtain the following result:

**Theorem 4.4.** *Let $\tilde{\phi}$ be the mapping defined in 3.6, and assume that all choices of $m(m+1)/2$ distinct rows of $X$ always have their mapping independent mappings for $\tilde{\phi}$. Then the outcome of the greedy algorithm initialized with the support of the solution to the continuous relaxation verifies*

$$f_\ell(S^+) \leq \log\left( \frac{k+m(m-1)/2+\ell}{k-m+1}\right) + f_\ell(S^*).$$

## 4.3 Computational considerations

Computing the $\ell$-th elementary symmetric polynomial on a vector of size $m$ can be done in $\mathcal{O}(m \log^2 \ell)$ using Fast Fourier Transform for polynomial multiplication, due to the construction introduced by Ben-Or (see [37]); hence, computing $f_\ell(S)$ requires $\mathcal{O}(nm^2)$ time, where the cost is dominated by computing $X_S^\top X_S$. Alg. 1 runs in expectation in $\mathcal{O}(n)$; Alg. 2 costs $\mathcal{O}(m^2 n^3)$.

# 5 Further Implications

We close our theoretical presentation by discussing a potentially important geometric problem related to ESP-design. In particular, our motivation here is the dual problem of D-optimal design (i.e., dual to the convex relaxation of D-optimal design): this is nothing but the well-known *Minimum Volume Covering Ellipsoid (MVCE)* problem, which is a problem of great interest to the optimization community in its own right—see the recent book [42] for an excellent account.

With this motivation, we develop the dual formulation for ESP-design now. We start by deriving $\nabla E_\ell(A)$, for which we recall that $E_\ell(\cdot)$ is a spectral function, whereby the spectral calculus of Lewis [29] becomes applicable, saving us from intractable multilinear algebra [23]. More precisely, say $U^\top \Lambda U$ is the eigendecomposition of $A$, with $U$ unitary. Then, as $E_\ell(A) = e_\ell \circ \lambda(A)$,

$$\nabla E_\ell(A) = U^\top \operatorname{Diag}(\nabla e_\ell(\Lambda))U = U^\top \operatorname{Diag}(e_{\ell-1}(\Lambda^{-(i)}))U. \tag{5.1}$$

We can now derive the dual of ESP-design (we consider only $z \geq 0$); in this case problem (3.2) is

$$\sup_{A \succ 0, z \geq 0} \inf_{\mu \in \mathbb{R}, H} -\tfrac{1}{\ell} \log E_\ell(A) - \operatorname{tr}(H(A^{-1} - X^\top \operatorname{Diag}(z)X)) - \mu(\mathbf{1}^\top z - k),$$

which admits as dual

$$\inf_{\mu \in \mathbb{R}, H} \sup_{A \succ 0, z \geq 0} \underbrace{-\tfrac{1}{\ell} \log E_\ell(A) - \operatorname{tr}(HA^{-1})}_{g(A)} + \operatorname{tr}(HX^\top \operatorname{Diag}(z)X) - \mu(\mathbf{1}^\top z - k). \tag{5.2}$$

We easily show that $H \succeq 0$ and that $g$ reaches its maximum on $\mathbb{S}_m^{++}$ for $A$ such that $\nabla g = 0$. Rewriting $A = U^\top \Lambda U$, we have

$$\nabla g(A) = 0 \iff \Lambda \operatorname{Diag}\big(e_{\ell-1}(\Lambda_{(i)})\big)\Lambda = e_\ell(\Lambda)UHU^\top.$$

In particular, $H$ and $A$ are co-diagonalizable, with $\Lambda \operatorname{Diag}(e_{\ell-1}(\Lambda_{(i)}))\Lambda = \operatorname{Diag}(h_1, \ldots, h_m)$. The eigenvalues of $A$ must thus satisfy the system of equations

$$\lambda_i^2 e_{\ell-1}(\lambda_1, \ldots, \lambda_{i-1}, \lambda_{i+1}, \ldots, \lambda_m) = h_i e_\ell(\lambda_1, \ldots, \lambda_m), \quad 1 \leq i \leq m.$$

Let $a(H)$ be such a matrix (notice, $a(H) = \nabla g^*(0)$). Since $f_\ell$ is convex, $g(a(H)) = f_\ell^\star(-H)$ where $f_\ell^\star$ is the Fenchel conjugate of $f_\ell$. Finally, the dual optimization problem is given by

$$\sup_{x_i^\top H x_i \leq 1, H \succeq 0} f_\ell^\star(-H) = \sup_{x_i^\top H x_i \leq 1, H \succeq 0} \tfrac{1}{\ell} \log E_\ell(a(H))$$

Details of the calculation are provided in Appendix D. In the general case, deriving $a(H)$ or even $E_\ell(a(H))$ does not admit a closed form that we know of. Nevertheless, we recover the well-known duals of A-optimal design and D-optimal design as special cases.

**Corollary 5.1.** *For $\ell = 1$, $a(H) = \operatorname{tr}(H^{1/2})H^{1/2}$ and for $\ell = m$, $a(H) = H$. Consequently, we recover the dual formulations of A- and D-optimal design.*

# 6 Experimental results

We compared the following methods to solving (3.1):

- UNIF / UNIFFDV: $k$ experiments are sampled uniformly / with Fedorov exchange
- GREEDY / GREEDYFDV: greedy algorithm (relaxed init.) / with Fedorov exchange
- SAMPLE: sampling (relaxed init.) as in Algorithm 1.

We also report the results for solution of the continuous relaxation (RELAX); the convex optimization was solved using projected gradient descent, the projection being done with the code from [12].

## 6.1 Synthetic experiments: optimization comparison

We generated the experimental matrix $X$ by sampling $n$ vectors of size $m$ from the multivariate Gaussian distribution of mean 0 and sparse precision $\Sigma^{-1}$ (density $d$ ranging from 0.3 to 0.9). Due to the runtime of Fedorov methods, results are reported for only one run; results averaged over multiple iterations (as well as for other distributions over $X$) are provided in Appendix E.

As shown in Fig. 1, the greedy algorithm applied to the convex relaxation's support outperforms sampling from the convex relaxation solution, and does as well as the usual Fedorov algorithm UNIFFDV; GREEDYFDV marginally improves upon the greedy algorithm and UNIFFDV. Strikingly, GREEDY provides designs of comparable quality to UNIFFDV; furthermore, as very few local exchanges improve upon its design, running the Fedorov algorithm with GREEDY initialization is much faster (Table 1); this is confirmed by Table 2, which shows the number of experiments in common for different algorithms: GREEDY and GREEDYFDV only differ on very few elements. As the budget $k$ increases, the difference in performances between SAMPLE, GREEDY and the continuous relaxation decreases, and the simpler SAMPLE algorithm becomes competitive. Table 3 reports the support of the continuous relaxation solution for ESP-design with $\ell = 10$.

Table 1: Runtimes (s) ($\ell = 10$, $d = 0.6$)

| $k$ | 40 | 80 | 120 | 160 | 200 |
|---|---|---|---|---|---|
| GREEDY | $2.8\ 10^1$ | $2.7\ 10^1$ | $3.1\ 10^1$ | $4.0\ 10^1$ | $5.2\ 10^1$ |
| GREEDYFDV | $6.6\ 10^1$ | $2.2\ 10^2$ | $3.2\ 10^2$ | $1.2\ 10^2$ | $1.3\ 10^2$ |
| UNIFFDV | $1.6\ 10^3$ | $4.1\ 10^3$ | $6.0\ 10^3$ | $6.2\ 10^3$ | $4.7\ 10^3$ |

Table 2: Common items between solutions ($\ell = 10$, $d = 0.6$)

| $k$ | 40 | 80 | 120 | 160 | 200 |
|---|---|---|---|---|---|
| \|GREEDY $\cap$ UNIFFDV\| | 26 | 76 | 114 | 155 | 200 |
| \|GREEDY $\cap$ GREEDYFDV\| | 40 | 78 | 117 | 160 | 200 |
| \|UNIFFDV $\cap$ GREEDYFDV\| | 26 | 75 | 113 | 155 | 200 |

Table 3: $\|z^*\|_0$ ($\ell = 10$, $d = 0.6$)

| $k$ | 40 | 80 | 120 | 160 | 200 |
|---|---|---|---|---|---|
| $d = 0.3$ | $93 \pm 3$ | $117 \pm 3$ | $148 \pm 2$ | $181 \pm 3$ | $213 \pm 2$ |
| $d = 0.6$ | $92 \pm 7$ | $117 \pm 4$ | $145 \pm 4$ | $180 \pm 3$ | $214 \pm 4$ |
| $d = 0.9$ | $88 \pm 3$ | $116 \pm 3$ | $147 \pm 4$ | $179 \pm 3$ | $214 \pm 1$ |

## 6.2 Real data

We used the Concrete Compressive Strength dataset [47] (with column normalization) from the UCI repository to evaluate ESP-design on real data; this dataset consists in 1030 possible experiments to model concrete compressive strength as a linear combination of 8 physical parameters. In Figure 2 (a), OED chose $k$ experiments to run to estimate $\theta$, and we report the normalized prediction error on the remaining $n - k$ experiments. The best choice of OED for this problem is of course A-optimal design, which shows the smallest predictive error. In Figure 2 (b), we report the fraction of non-zero entries in the design matrix $X_S$; higher values of $\ell$ correspond to increasing sparsity. This confirms that OED allows us to scale between the extremes of A-optimal design and D-optimal design to tune desirable side-effects of the design; for example, sparsity in a design matrix can indicate not needing to tune a potentially expensive experimental parameter, which is instead left at its default value.

## 7 Conclusion and future work

We introduced the family of ESP-design problems, which evaluate the quality of an experimental design using elementary symmetric polynomials, and showed that typical approaches to optimal design such as continuous relaxation and greedy algorithms can be extended to this broad family of problems, which covers A-optimal design and D-optimal design as special cases.

We derived new properties of elementary symmetric polynomials: we showed that they are geodesically log-convex on the space of positive definite matrices, enabling fast solutions to solving the relaxed ESP optimization problem. We furthermore showed in Lemma 4.2 that volume sampling, applied to the columns of the design matrix $X$ has a constant multiplicative impact on the objective function $E_\ell\big(\big(X_S^\top X_S\big)^{-1}\big)$, extending Avron and Boutsidis [2]'s result from the trace to all el-

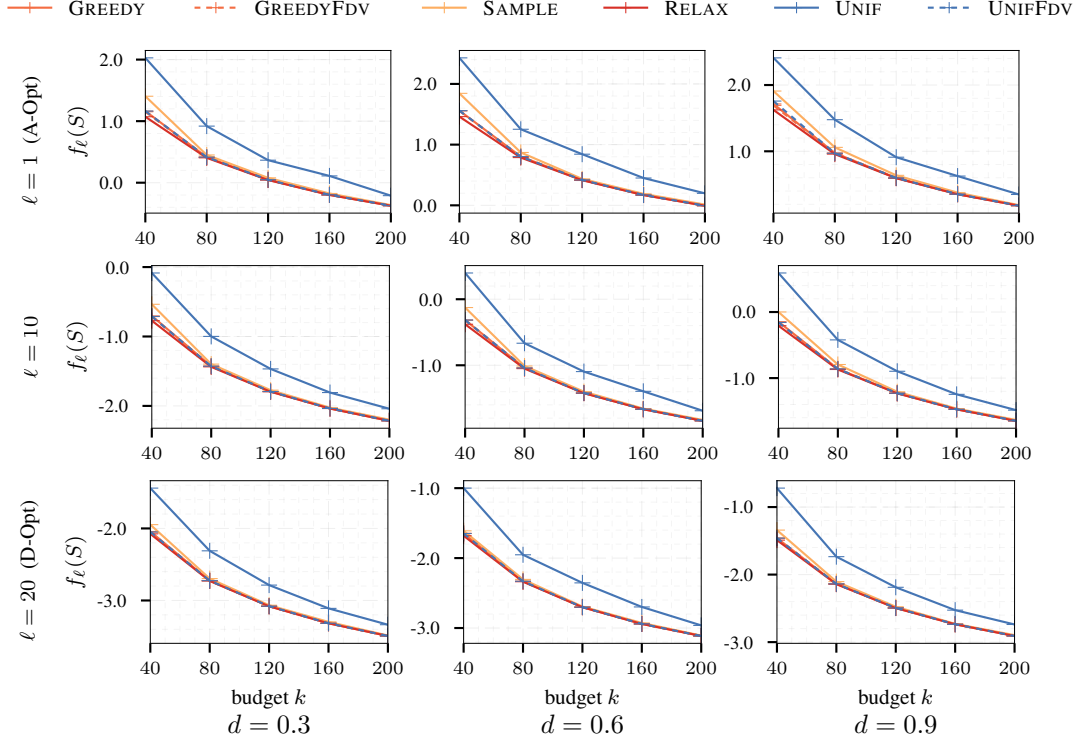

Figure 1: Synthetic experiments, $n = 500$, $m = 30$. The greedy algorithm performs as well as the classical Fedorov approach; as $k$ increases, all designs except UNIF converge towards the continuous relaxation, making SAMPLE the best approach for large designs.

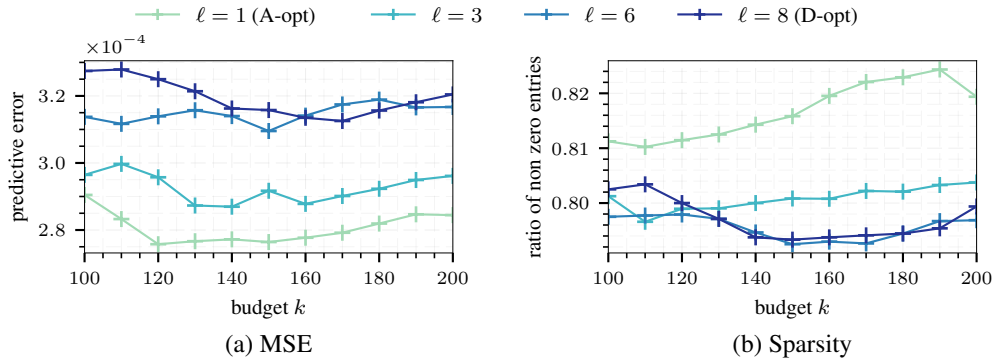

Figure 2: Predicting concrete compressive strength via the greedy method; higher $\ell$ increases the sparsity of the design matrix $X_S$, at the cost of marginally decreasing predictive performance.

ementary symmetric polynomials. This allows us to derive a greedy algorithm with performance guarantees, which empirically performs as well as Fedorov exchange, in a fraction of the runtime.

However, our work still includes some open questions: in deriving the Lagrangian dual of the optimization problem, we had to introduce the function $a(H)$ which maps $\mathbb{S}_m^{++}$; however, although $a(H)$ is known for $\ell = 1, m$, its form for other values of $\ell$ is unknown, making the dual form a purely theoretical object in the general case. Whether the closed form of $a$ can be derived, or whether $E_\ell(a(H))$ can be obtained with only knowledge of $H$, remains an open problem. Due to the importance of the dual form of D-optimal design as the Minimum Volume Covering Ellipsoid, we believe that further investigation of the general dual form of ESP-design will provide valuable insight, both into optimal design and for the general theory of optimization.

ACKNOWLEDGEMENTS

Suvrit Sra acknowledges support from NSF grant IIS-1409802 and DARPA Fundamental Limits of Learning grant W911NF-16-1-0551.

## Footnotes

[1]For a more in depth discussion of the geometric interpretation of various optimal designs, refer to e.g. [7, Section 7.5].

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
