[Supplementary Material]

# A Geodesic convexity

Recall that we are using the notation

$$P \#_t Q := P^{1/2}(P^{-1/2}QP^{-1/2})^t P^{1/2}, \quad t \in [0,1], \text{ and } P, Q \succ 0,$$

to denote the *geodesic* between positive definite matrices $P$ and $Q$ under the Riemannian metric $g_P(X, Y) = \operatorname{tr}(P^{-1}XP^{-1}Y)$. The midpoint of this geodesic is $P \#_{1/2} Q$, and it is customary to drop the subscript and just write $P \# Q$.

## A.1 Proof of Lemma 3.3

Here we prove the log-g-convexity of $E_\ell$ on the set of psd matrices. As far as we are aware, this result is novel. By continuity, it suffices to prove midpoint log-g-convexity; that is, it suffices to prove

$$E_\ell(P \# Q) \le \sqrt{E_\ell(P)E_\ell(Q)}.$$

From basic multilinear algebra (see e.g., [5, Ch. 1]) we know that for any $n \times n$ matrix $P$, there exists a projection matrix $W$ such that $E_\ell(P) = \operatorname{tr} \wedge^\ell P = \operatorname{tr} W^* P^{\otimes n} W$. [40, Lemma 2.23] shows that

$$(P \# Q)^{\otimes n} = P^{\otimes n} \# Q^{\otimes n}.$$

Thus, it follows that

$$
\begin{aligned}
E_\ell(P \# Q) &= \operatorname{tr} W^* (P \# Q)^{\otimes n} W = \operatorname{tr} W^* [P^{\otimes n} \# Q^{\otimes n}] W \\
&\le [\operatorname{tr} W^* P^{\otimes n} W]^{1/2} [\operatorname{tr} W^* Q^{\otimes n} W]^{1/2} \\
&= [E_\ell(P) E_\ell(Q)]^{1/2},
\end{aligned}
$$

where the inequality follows from log-g-convexity of the trace map [40, Cor. 2.9]. $\qquad\square$

Observe that this result is *stronger* than the usual log-convexity result, which it yields as a corollary.

## A.2 Proof of Corollary 3.4

We present now a short new proof of the log-convexity of the map $Z \mapsto E_\ell(A^\top Z A)^{-1}$; we assume that $A$ has full column rank. As before, it suffices to prove midpoint convexity. Let $Z, Y \succ 0$. We must then show that

$$\log E_\ell \left( A^\top \left( \tfrac{Z+Y}{2} \right) A \right)^{-1} \le \tfrac{1}{2} \log E_\ell(A^\top Z A)^{-1} + \tfrac{1}{2} \log E_\ell(A^\top Y A)^{-1}.$$

Since (e.g., [6, Ch. 5]) $(A^\top Z A) \# (A^\top Y A) \le \frac{A^\top Z A + A^\top Y A}{2}$, we get $\left[ A^\top \left( \tfrac{Z+Y}{2} \right) A \right]^{-1} \le [(A^\top Z A) \# (A^\top Y A)]^{-1}$. Since $\log E_\ell$ is monotonic in Löwner order (Prop. 2.1-(i)), we see that

$$\log E_\ell \left( A^\top \left( \tfrac{Z+Y}{2} \right) A \right)^{-1} \le \log E_\ell \left( [(A^\top Z A) \# (A^\top Y A)]^{-1} \right) = \log E_\ell [(A^\top Z A)^{-1} \# (A^\top Y A)^{-1}].$$

But from Lemma 3.3 we know that $E_\ell(P \# Q) \le \sqrt{E_\ell(P)E_\ell(Q)}$, which allows us to write

$$\log E_\ell [(A^\top Z A)^{-1} \# (A^\top Y A)^{-1}] \le \tfrac{1}{2} E_\ell(A^\top Z A)^{-1} + \tfrac{1}{2} E_\ell(A^\top Y A)^{-1},$$

which completes the proof. $\qquad\square$

# B Bounding the support of the continuous relaxation

As mentioned in the main paper, this proof is identical to the proof provided by [44, Lemma 3.5] for A-optimal design once we derive $\nabla f_\ell(z)$; we reproduce it here for completeness.

*Proof. (Theorem 3.6).* It is easy to show from (5.1) and Prop. 2.1-*(iii)* that

$$\frac{\partial f_\ell(z)}{\partial z_i} = -\frac{1}{\ell} x_i^\top \underbrace{U \left( \Lambda^{-1} - \nabla e_{m-l}(\Lambda)/e_{m-\ell}(\Lambda) \right) U^\top}_{W} x_i$$

and that $W$ is positive definite.

Assume now that all choices of $m(m+1)/2$ distinct rows of $X$ have their mapping under $\tilde{\phi}$ be independent. We now consider the Lagrangian multiplier version of (3.2):

$$f(z, u_i, v_i, \lambda) = f_\ell(z) - \sum_i u_i z_i + \sum_i (z_i - 1) + \lambda(\sum_i z_i - k)$$

Let $z^*$ be the optimal solution, and let $A \subseteq [n]$ be the indices $i$ such that $0 < z_i < 1$. Assume by contradiction that $|A| > m(m+1)/2$. By KKT conditions, we have for $i \in A$,

$$-\frac{\partial f(z^*)}{\partial z_i} = x_i^\top W x_i = \langle \phi(x) \mid \psi(W) \rangle = \mu \tag{B.1}$$

where $\phi$ is the mapping defined in Theorem 3.6 and $\psi$ takes the upper triangle of a symmetric matrix and maps it to a vector of size $m(m+1)/2$. Then, (B.1) can be rewritten for $m(m+1)/2$ indices in $A$ as the following linear system of variables:

$$\begin{pmatrix} \tilde{\phi}(x_1) \\ \dots \\ \tilde{\phi}(x_{m(m+1)/2+1}) \end{pmatrix} \begin{pmatrix} \psi(W) \\ -\lambda \end{pmatrix} = 0. \tag{B.2}$$

By hypothesis, the first matrix is invertible and hence $\psi(W)$ and $\lambda$ must be 0, which contradicts the strict positive definiteness of $W$. $\qquad\square$

## C  Greedy algorithm details

To analyze our greedy algorithm, we need the following lemma, which is an extension of [2, Lemma 3.9] to all elementary symmetric polynomials:

**Lemma C.1.** *Let $X \in \mathbb{R}^{n \times m}(n \geq m)$ be a matrix with full column rank, and let $k$ be a budget $m \leq k \leq n$. Let $S$ be a random variable with probability*

$$P_S = \frac{\det(X_S^\top X_S)}{\sum_{T \subseteq [n], |T|=k} \det(X_T^\top X_T)}.$$

*Then*

$$\mathbb{E}\left[ E_\ell \left( (X_S^\top X_S)^{-1} \right) \right] \leq \left( \prod_{i=1}^\ell \frac{n-m+i}{k-m+i} \right) E_\ell \left( (X^\top X)^{-1} \right). \tag{C.1}$$

*Proof.* The below calculations depend heavily on the Cauchy-Binet formula, of which we reproduce a special case here for $X \in \mathbb{R}^{n \times m}$:

$$\det(X^\top X) = \sum_{S \subseteq [n], |S|=m} \det(X_S^\top X_S). \tag{C.2}$$

We also use the representation (2.2). By definition we have

$$\mathbb{E}\left[ E_\ell \left( (X_S^\top X_S)^{-1} \right) \right] = \frac{\sum_{S \subseteq [n], |S|=k} \det(X_S^\top X_S) E_\ell \left( (X_S^\top X_S)^{-1} \right)}{\sum_{S \subseteq [n], |S|=k} \det(X_S^\top X_S)}$$

For the denominator, we have

$$\sum_{S \subseteq [n], |S|=k} \det(X_S^\top X_S) = \sum_{S \subseteq [n], |S|=k} \det(X_S^\top X_S)$$

$$\stackrel{(a)}{=} \sum_{S \subseteq [n], |S|=k} \sum_{T \subseteq S, |T|=m} \det(X_T^\top X_T)$$

$$\stackrel{(b)}{=} \binom{n-m}{k-m} \sum_{T \subseteq S, |T|=m} \det(X_T^\top X_T)$$

$$\stackrel{(c)}{=} \binom{n-m}{k-m} \det(X^\top X)$$

where $(a)$ is obtained using the Cauchy-Binet formula (C.2), $(b)$ by noticing that there are $\binom{n-m}{k-m}$ sets of size $k$ that contain a set $T$ of size $m$, and $(c)$ by reapplying (C.2).

For the numerator, we first use the fact that $E_\ell(A^{-1}) = \frac{1}{\det A} E_{m-\ell}(A)$:

$$
\sum_{S\subseteq[n],|S|=k} \det(X_S^\top X_S) E_\ell\left((X_S^\top X_S)^{-1}\right) \overset{(a)}{\leq} \sum_{S\subseteq[n],|S|=k} E_{m-\ell}(X_S^\top X_S)
$$

$$
= \sum_{S\subseteq[n],|S|=k}\ \sum_{L\subseteq[m],|L|=m-\ell} (X_S^\top X_S)[L|L]
$$

$$
\overset{(b)}{=} \sum_{S\subseteq[n],|S|=k}\ \sum_{L\subseteq[m],|L|=m-\ell} \det((Y_L)_S^\top (Y_L)_S)
$$

$$
\overset{(c)}{=} \sum_{S\subseteq[n],|S|=k}\ \sum_{L\subseteq[m],|L|=m-\ell}\ \sum_{T\subseteq S,|T|=m-\ell} \det((Y_L)_T^\top (Y_L)_T)
$$

$$
= \binom{n-m+\ell}{k-m+\ell} \sum_{L\subseteq[m],|L|=m-\ell}\ \sum_{T\in[n],|T|=m-\ell} \det((Y_L)_T^\top (Y_L)_T)
$$

$$
\overset{(d)}{=} \binom{n-m+\ell}{k-m+\ell} \sum_{L\subseteq[m],|L|=m-\ell} \det((Y_L)^\top (Y_L))
$$

$$
= \binom{n-m+\ell}{k-m+\ell} \sum_{L\subseteq[m],|L|=m-\ell} (X^\top X)[L|L]
$$

$$
= \binom{n-m+\ell}{k-m+\ell} E_{m-\ell}(X^\top X)
$$

Here, $(a)$ is just (2.2); we have equality if all subsets $S$ of size $k$ produce strictly positive definite matrices $X_S^\top X_S$. For $(b)$, we note $Y_L$ the submatrix of $X$ with all columns but those in $L$ removed; then, $(Y_L)_S^\top (Y_L)_S = [X_S^\top X_S]$ for all subsets $S$. $(d)$ is an application of Cauchy-Binet. Hence,

$$
\mathbb{E}\left[E_\ell\left((X_S^\top X_S)^{-1}\right)\right] = \frac{\sum_{S\subseteq[n],|S|=k} \det(X_S^\top X_S) E_\ell\left((X_S^\top X_S)^{-1}\right)}{\sum_{S\subseteq[n],|S|=k} \det(X_S^\top X_S)}
$$

$$
= \frac{\binom{n-m+\ell}{k-m+\ell} E_{m-\ell}(X^\top X)}{\binom{k-m}{n-m} \det(X^\top X)}
$$

$$
= \left(\prod_{i=1}^{\ell} \frac{n-m+i}{k-m+i}\right) E_\ell((X^\top X)^{-1})
$$

$\square$

We can now prove Theorem 4.3:

*Proof.* We recursively show that greedily removing $j$ items constructs a set $S$ (of size $(n-j)$) s.t.

$$
E_\ell\left((X_S^\top X_S)^{-1}\right) \leq \left(\prod_{i=1}^{\ell} \frac{n-m+i}{n-j-m+i}\right) E_\ell\left((X^\top X)^{-1}\right). \tag{C.3}
$$

(C.3) is trivially true for $j=0$. Assume now that (C.3) holds for $j \geq 0$, and let $S_j$ be the corresponding set of size $(n-j)$. Let now $S_{j+1}$ be the set of size $|S_j|-1$ that minimizes $E_\ell(X_{S_{j+1}}^T X_{S_{j+1}})$.

From lemma 4.2, we know that for sets $S$ of size $|S_j|-1$ drawn according to dual volume sampling,

$$
\mathbb{E}\left[E_\ell\left((X_S^\top X_S)^{-1}\right)\right] \leq \left(\prod_{i=1}^{\ell} \frac{|S_j|-m+i}{(|S_j|-1)-m+i}\right) E_\ell\left((X_{S_j}^\top X_{S_j})^{-1}\right).
$$

In particular, the minimum of $E_\ell(X_S^T X_S)$ over all sets of size $|S_j| - 1$ is upper bounded by the expectancy: $E_\ell(X_{S_{j+1}}^T X_{S_{j+1}}) \leq \left( \prod_{i=1}^{\ell} \frac{n-j-m+i}{n-j-1-m+i} \right) E_\ell \left( \left( X_{S_j}^\top X_{S_j} \right)^{-1} \right)$.

By recursion hypothesis applied to $S_j$, we then have

$$E_\ell(X_{S_{j+1}}^T X_{S_{j+1}}) \leq \left( \prod_{i=1}^{\ell} \frac{n-j-m+i}{n-j-1-m+i} \right) E_\ell \left( \left( X_{S_j}^\top X_{S_j} \right)^{-1} \right)$$

$$\leq \left( \prod_{i=1}^{\ell} \frac{n-j-m+i}{n-j-1-m+i} \right) \left( \prod_{i=1}^{\ell} \frac{n-m+i}{n-j-m+i} \right) E_\ell \left( (X^\top X)^{-1} \right)$$

$$\leq \left( \prod_{i=1}^{\ell} \frac{n-m+i}{n-(j+1)-m+i} \right) E_\ell \left( (X^\top X)^{-1} \right),$$

which concludes the recursion. Then, constructing a set of size $k$ amounts to setting $j = n - k$ in Eq.(C.3), which proves Eq. (4.2). $\qquad\square$

## D   Obtaining the dual formulation

We first show that (5.2) has $H \succ 0$: by contradiction, assume that there exists $x$ such that $x^\top H x < 0$ and $\|x\| = 1$. Then setting $A = I - \frac{t}{1+t} x x^\top$ has $g(A)$ go to infinity with $t$.

Next, $g(A) = -\frac{1}{\ell} \log E_\ell(A) - \text{tr}(HA^{-1})$ reaches its maximum on $\mathbb{S}_m^{++}$: if $\|A\| \to \infty$, we easily have $g \to -\infty$. The same holds for $A \to \partial \mathbb{S}_m^{++}$.

We now derive the dual form:

$$(5.2) \iff \inf_{\substack{\mu \in \mathbb{R}, \\ H \in \mathbb{R}^{m \times m}}} \sup_{A \succ 0, z \geq 0} -\frac{1}{\ell} \log E_\ell(A) - \text{tr}(HA^{-1}) + \text{tr}(HX^\top \text{Diag}(z)X) - \mu(\mathbf{1}^\top z - k)$$

$$(5.2) \iff \inf_{\mu \in \mathbb{R}, H \succeq 0} \left[ f_\ell^\star(-H) + \sup_{z \geq 0} \text{tr}(HX^\top \text{Diag}(z)X) - \mu(\mathbf{1}^\top z - k) \right]$$

$$(5.2) \iff \inf_{\mu \in \mathbb{R}, H \succeq 0} \left[ f_\ell^\star(-H) + \sup_{z \geq 0} \sum_i z_i (x_i^\top H x_i - \mu) + \mu k \right]$$

$$(5.2) \iff \inf_{\substack{x_i^\top H x_i \leq \mu, \\ H \succeq 0}} f_\ell^\star(-H) + k\mu$$

$$(5.2) \iff \sup_{\substack{x_i^\top H x_i \leq 1, \\ H \succeq 0, \mu > 0}} -f_\ell^\star(-\mu H) - k\mu$$

$$(5.2) \stackrel{\star}{\iff} \sup_{\substack{x_i^\top H x_i \leq 1, \\ H \succeq 0}} -f_\ell^\star(-H)$$

Where $\stackrel{\star}{\iff}$ follows from $f_\ell^\star(-\mu H) = \sup_{A \succ 0} -\text{tr}(HA) - f_\ell(A/\mu) = f_\ell^\star(-H) - \log \mu$.

Finally, we saw that by definition of $a(H)$, $f_\ell^\star(-H) = g(a(H)) = -E_\ell(a(H)) - \text{tr}(H (a(H))^{-1})$, and that the eigenvalues $\Lambda$ of $a(H)$ verify

$$\lambda_i^2 \frac{e_{\ell-1}(\lambda_1, \ldots, \lambda_{i-1}, \lambda_{i+1}, \ldots, \lambda_m)}{e_\ell(\lambda_1, \ldots, \lambda_m)} = h_i, \quad 1 \leq i \leq m.$$

Then,

$$e_\ell(\lambda_1,\ldots,\lambda_m)\operatorname{tr}(H\,(a(H))^{-1}) = \sum_i \frac{1}{\lambda_i}\lambda_i^2 e_{\ell-1}(\lambda_1,\ldots,\lambda_{i-1},\lambda_{i+1},\ldots,\lambda_m)$$

$$= \sum_i \lambda_i e_{\ell-1}(\lambda_1,\ldots,\lambda_{i-1},\lambda_{i+1},\ldots,\lambda_m)$$

$$= \sum_i \sum_{J\subseteq[n],|J|=\ell,i\in J}\prod_{j\in J}\lambda_j$$

Each subset $J$ is hence going to appear $\ell$ times, once for each of its elements; finally

$$\operatorname{tr}(H\,(a(H))^{-1}) = \ell \sum_{J\subseteq[n],|J|=\ell}\prod_{j\in J}\lambda_j/e_\ell(\lambda_1,\ldots,\lambda_m) = \ell$$

and hence

$$\sup_{\substack{x_i^\top H x_i \le 1,\\ H\succeq 0}} -f_\ell^\star(-H) \iff \sup_{\substack{x_i^\top H x_i \le 1,\\ H\succeq 0}} -g(a(H)) = \sup_{\substack{x_i^\top H x_i \le 1,\\ H\succeq 0}} \frac{1}{\ell}\log E_\ell(a(H)) + \ell.$$

## E  Additional synthetic experimental results

To compare to [44], we generated the experimental matrix $X$ by sampling $n$ vectors of size $m$ from the multivariate Gaussian distribution of mean 0 and covariance $\Sigma = \operatorname{Diag}(1^{-\alpha},\ldots,m^{-\alpha})$ for various sizes of $\alpha$ and multiple budgets $k$, with $m = 50, n = 1000$; $\alpha$ controls hows *skewed* the distribution is.

Table 4: $\|z\|_0$ for $n = 500$, $m = 30$, $\ell = 15$

|              | $k = 60$   | $k = 120$  | $k = 180$  | $k = 240$  | $k = 300$  |
|--------------|------------|------------|------------|------------|------------|
| $\alpha = 1$ | $167 \pm 9$ | $192 \pm 6$ | $241 \pm 5$ | $290 \pm 4$ | $335 \pm 4$ |
| $\alpha = 2$ | $160 \pm 4$ | $187 \pm 5$ | $240 \pm 2$ | $284 \pm 3$ | $331 \pm 6$ |
| $\alpha = 3$ | $160 \pm 4$ | $190 \pm 3$ | $237 \pm 5$ | $281 \pm 4$ | $333 \pm 3$ |

Figure 3: Synthetic experiments, $n = 500$, $m = 30$, $\ell = 1$.

Figure 4: Synthetic experiments, $n = 500$, $m = 30$, $\ell = 10$.

Figure 5: Synthetic experiments, $n = 500$, $m = 30$, $\ell = 30$.

Figure 6: Synthetic experiments, $n = 500$, $m = 30$, $\ell = 1$.

Figure 7: Synthetic experiments, $n = 500$, $m = 30$, $\ell = 15$.

Figure 8: Synthetic experiments, $n = 500$, $m = 30$, $\ell = 30$.