[Reviews · NeurIPS 2017]

Reviewer 1



The authors present new results in the area of optimal experimental design (OED). My main concern about the paper is that the writing is very dense and appears to require a thorough understanding of OED. I am not sure that fits well with the audience for NIPS. So, at best the paper would be targeting a small subset of the audience. My question is whether there is a way to make the paper better suited for a broader audience. (E.g., a more detailed discussion of the setup and results on real data could help introduce key concepts from OED.) Based on author feedback and the other reviewers' comments, it appears the results will be of interest to a subset of the NIPS audience. I'm therefore revising my score. I would still encourage the authors to write a few intro paragraphs to draw in a broader audience.

Reviewer 2



The authors present a novel objective for experiment design that interpolates A- and D-optimality. They present a convex relaxation for the problem which includes novel results on elementary symmetric polynomials, and an efficient greedy algorithm based on the convex relaxation with performance guarantees for the original task. They present results on synthetic data, and present results on a concrete compressive strength task. I am not an expert in this area but I believe I understand the contribution and that it is significant. The paper is extremely well-presented, with minor points of clarification needed that I have identified below. The main suggestion I have is that the authors more explicitly explain the benefits of the new method in terms of the properties of solutions that interpolate between A and D -- In 6.2, the authors present the ability to use their method to get a spars(er) design matrix as an important feature. This should be stated at the beginning of the paper if it is a true aim (and not simply an interesting side-effect) and the authors should review other methods for obtaining a sparse design matrix. (I presume that there are e.g. L1 penalized methods that can achieve sparsity in this area but I do not know.) In general, a more explicit statement of the practical benefits of the contributions in the paper would be useful for the reader. l21. is a bit confusing regarding use of "vector" -- \epsilon_i is a scalar, yes? l22. Gauss-Markov doesn't require \epsilon_i be Gaussian but it *does* require homoskedasticity. Clarify. l45. "deeply study" - studied l77. I think (2.1) is confusing with v as vectors unless the product is element-wise. Is this intended? l107. "Clearly..." - Make a more precise statement. Is an obvious special case hard? What problem of known hardness reduces to this problem? If you mean "no method for finding an exact solution is known" or "we do not believe an efficient exact solution is possible" then say that.

Reviewer 3



The paper proposes more general formulations of optimal experiment design. More specifically, given a set of feature vectors, optimal experiment design tries to choose which of these to get labels for so that the variance of the learned model can be minimized. While previous approaches have tried to minimize the trace (A-optimality) or the determinant (D-optimality) of the covariance matrix, in this paper, the authors propose a more general formulation whose extremes include these two cases. Based on a convex formulation of this problem, the authors then go on to propose 2 algorithms to perform optimal experiment design. The connection between elementary symmetric polynomials and its special cases was an interesting revelation. I am not enough of an expert in the area to know if this connection has been made previously or in other contexts. In the paper, the authors introduce this notion in a very matter-of-fact kind of way, without much fanfare (I had difficulty opening the supplementary material pdf which looked blank). The reduction to the elegant log det minimization immediately lets the authors utilize convex relaxations. As is pointed out in the paper, the algorithms proposed in the paper have a lot of similarity with the ones for trace minimization in reference 44. One issue/question I had is whether these algorithms are of practical importance since I haven't seen much use of A-optimality or D-optimality in the field, and now with this generalization, the practitioner will need to figure the value of the extra l parameter (this question is not addressed in the paper). Another point that may improve the paper is to report errors from models learned without choosing the rows to show whether we are losing any accuracy by minimizing the number of examples used for training.